# Correlation between the DJSI Chile and the Financial Indices of Chilean Companies

**Karime Chahuán-Jiménez** 

Escuela de Auditoría, Centro de Investigación en Negocios y Gestión Empresarial, Universidad de Valparaíso, Valparaíso 2340000, Chile; karime.chahuan@uv.cl; Tel.: +56-99-289-5030

**Abstract:** The Dow Jones Sustainability Index Chile (DJSI Chile) is made up of leading sustainability companies that are investing great effort into sustainable management. This study correlates the DJSI Chile with the financial indices (return on equity (ROE), return on assets (ROA), market value, earnings, and leverage) of companies that belong to the General Stock Price Index (IGPA) in Chile. The methodology used was quantitative, considering Chilean companies in the IGPA, including companies belonging to the DJSI Chile, applying a normality and correlation test based on the results. In conclusion, the study shows that in the results for the ROE, ROA, and leverage variables, there is no positive correlation with the DJSI Chile. However, the DJSI Chile is correlated with market value (for approximately 80% of the companies), and with earnings, there is a slightly higher correlation for the companies that belong to the DJSI Chile than for the remaining companies in the IGPA, thus if there exists a correlation between the DJSI Chile index and the variables market value and earnings, the index enables the prediction of those financial variables or predicts the finance indices (value market and earnings) of the companies that make up the DJSI Chile basing in the DJSI Chile index.

**Keywords:** integrated reporting; sustainability reporting; DJSI Chile; financial indices; financial markets

**JEL Classification:** G14; G49; D53

## 1. Introduction

The evolution of corporate social responsibility described by Soundararajan et al. (2018) began in 1972, and it corresponded to activities that could be considered social activities. This evolution continued until the year 2000 when a strategic management orientation was implemented in companies, and corporate social responsibility (CSR) emerged as an excellent indicator of the legitimacy of a firm. According to Rivera et al. (2019), CSR is understood as the extent to which firms assume economic, legal, ethical, social, and discretionary responsibilities vis à vis their stakeholders.

According to Chang et al. (2015), CSR in different industries has received a great deal of attention in recent years. As such, rating indices have been developed for evaluating the CSR performance of corporations in different focus environments, such as social and government.

Alzboun et al. (2016) indicated that sustainable business practices do not reduce the level of financial results. However, it is expected that financial results will be reduced through sustainability practices over time. This expectation conflicts with Yu and Zhao (2015) finding that results are consistent with the theory of value improvement with respect to the role of pursuing sustainability in the valuation of a company. In addition, the positive impact generated by the incorporation of sustainable practices in the value of a company, mainly in countries that have strong protection for investors and high levels of disclosure, complements greater financial transparency.

In addition to systemic risk, the results of research on the European DJSI have provided insights into the excess return that was positive and negative but statistically insignificant in other DJSI specifications (Sokolovska and Keseljevic 2019; López et al. 2007). Variations in sustainability indices differ from those of the main market indices.

The General Stock Price Index (IGPA) in Chile is composed of 83 Chilean companies, and the Dow Jones Sustainability Index (DJSI) Chile consists of 29 leading sustainability companies, determined by the total sustainability score (TTS), which identifies RobecoSAM (RobecoSam 2020) as the leading sustainability company based on its annual corporate sustainability assessment (CSA) (Lee et al. 2020; Searcy and Elkhawas 2012). In the case of Chile, these companies are also part of the IGPA. The index uses the best-in-class approach, which represents the top 40% of the IGPA of the Santiago Stock Exchange based on long-term environmental, social, and governance factors (S&P 2020).

The purpose of the research is to examine the correlation between the companies in the DJSI Chile, all of which are part of the IGPA, with the financial indices of the companies under study (Return on Equity (ROE), Return on assets (ROA), market value, earnings, and leverage). Companies belonging to the DJSI Chile, according to the theory, correlate with the financial variable indices. Therefore, variability in an index would be an indication of the financial conditions of the companies belonging to said index, thereby differentiating them from the remaining companies in the IGPA.

The research includes an analysis based on the normality testing (in this case, the Shapiro–Wilk test), which was chosen for the amount of data per company generated per year from the start of the DJSI Chile (fewer than 50 data points), and then it applies a correlation testing depending on the results of the normality test. The index selects the companies that combine economic success with sustainable development from the total number of companies that are incorporated into the IGPA.

Therefore, two research hypotheses are presented. The first hypothesis corresponds to the concept that the financial indices of the companies are correlated in the DJSI Chile, especially the companies that are part of this index. The second hypothesis is that the companies that are part of the DJSI Chile have a greater correlation with the financial indices than the companies that are part of the IGPA and that are not considered in the DJSI Chile because the former have a greater investment in and concern for sustainability.

The research is composed of the methodology, literature review, and comparative statistical analysis associated with the correlation of the variables for companies that are part of the DJSI Chile and the IGPA, as well the conclusions, limitations, and projections generated by the research.

## 2. Literature Review

From a financial point of view, in their meta-analysis, Orlitzky et al. (2003) showed that there is a positive association between corporate social performance (CSP) and corporate financial performance (CFP) across industries and across study contexts. Thus, they confirm conclusions, based on event studies, that support the validity of enlightened self-interest in social responsibility. According to Hawn et al. (2018), the valuation of sustainability by investors worldwide has evolved over time, implying a decrease in reactions to efforts by US companies.

The results of the research of Cunha and Moneva (2018) indicate that compliance with sustainability becomes an instrument of responsibility for stakeholders, so the transparency and legitimization of their activities are the main factors that influence their publications. This finding aligns with that of Grewal et al. (2020), who found that firms voluntarily disclosing more Sustainability Accounting Standards Board (SASB)-identified sustainability information exhibit greater price informativeness. Moreover, company legitimacy implies that companies modify their actions so that they are acceptable to the community and act in ways consistent with social values. Additionally, institutional investors have shown increasing interest in how companies align their CSR strategies with sustainable development goals (SDGs) according to Garcia-Sanchez et al. (2020).

For Marano and Kostova (2016), complex environments affect a company's adoption of CSR practices. To capture the effect of transnational fields, this study considers the institutional influences

of all of the environments of the countries to which a company is linked through its portfolio of operations; a set of factors that render certain pressures more prominent are identified, including the economic dependence of the company in a particular country, the heterogeneity of the institutional forces within the transnational field of the company, the exposure to leading countries with stricter CSR templates, and the intensity and commitment of economic links in particular (i.e., foreign direct investment versus international trade).

In addition, the study by Mura et al. (2018) revealed that sustainability measurements could inform the current debates about performance measurement and management in three main ways: by emphasizing the roles of those interested in the design, implementation, and use of the measures; by indicating ways in which to establish common measures and the sharing of data between organizations; and by adopting new theoretical perspectives. This outcome implies a link with aspects associated with the strategic performance of an organization and its development practices through the measurement of standards linked to social responsibility, as supported by the research of Awaysheh et al. (2020), indicating that best-in-class firms outperform their industry peers in terms of operating performance and have higher relative market valuations.

Considering the variables affected by the implementation of sustainable practices, Charlo et al. (2015) conclude that greater benefits are obtained for the same level of risk and they observed greater sensitivity to changes in the market, leverage levels and company size, and participation in institutional CSR activities—those aimed at a firm's secondary stakeholders or society at large providing an "insurance-like" benefit (Godfrey et al. 2009). A business that is proactive with respect to issues related to the environment can enjoy potential benefits such as improvements in corporate image and popularity, increased quality levels, recycling and pollution control, reduction in costs through energy conservation, improvement in relations with neighbors, and the creation of high-value green products (Chen et al. 2018). According to Lins et al. (2017), the trust between a firm and both its stakeholders and investors, built through investments in social capital, pays off when the overall level of trust in corporations and markets suffers a negative shock.

According to Lameira et al. (2013), there is a relationship between better sustainability practices and better development in companies; from an internal perspective, better sustainability practices bring about low risk (lower capital constraints) (Cheng et al. 2014) and high value in companies. Additionally, they indicate that market value, leverage, return on assets, and volatility are possible determinants of behavior practices, and in the analysis of their variables, they incorporate risk as an additional variable.

Hoi et al. (2018) presented a series of indices that could vary depending on the implementation of sustainability in companies, including social capital, company size, profitability of assets, leverage, risk variables, visibility of demand, policies implemented, and minority shareholders, among others.

The theoretical review presented by Lameira et al. (2013) and the studies of Yuan et al. (2020) suggest that performance is positively correlated with the improvement of practices, and Fu et al. (2020) reveal that the chief sustainability officer (CSO) increases the firm's socially responsible activities (CSR) and reduces its socially irresponsible activities (CSR). However, it is impossible to indicate the result of each variable since test errors can occur and are not conclusive, and the benefits of behavior can be achieved in several different dimensions and ultimately indicate that market value, operating leverage, return on assets, and volatility are possible determinants of the quality of sustainable practices. According to McWilliams and Siegel (2000), when the model is properly specified, it finds that CSR has a neutral impact on financial performance. This finding is complemented by Calveras and Ganuza (2018), who demonstrate that CSR can serve as a tool for a company's product differentiation strategy, finding that both internal and external CSR improve the product quality of a company and therefore improve the market vision of the company's reputation.

Yu and Zhao (2015) observe a positive relationship between the implementation of sustainability and company value. In the analysis, they incorporated the following variables: sustainability value, ROA, ROE, earnings per share, book value/market share value, book value/EBITDA (earnings before interest,

taxes, depreciation, and amortization), share price/profit, volatility, company cost value, and stock value (beta). The same authors indicate that in the case of developing countries, the DJSI includes companies across all industries that exceed their peers in sustainability measures, such as corporate governance, measures related to the environment, and measures related to relationships with stakeholders, with a special focus on specific sectors of the industry, based on a deep economic, environmental and social criteria analysis (Arribas et al. 2019). According to Waddock and Graves (1997), CSP is found to be positively associated with prior financial performance, supporting the theory that slack resource availability and CSP are positively related. CSP is also found to be positively associated with future financial performance, supporting the theory that good management and CSP are positively related.

Charlo et al. (2015) state that the variables associated with the market price are not associated with engaging in more socially responsible actions. Additionally, regarding the idea that social impact investment can improve portfolio risk-return performance, the results of our macro-asset allocation analysis show the importance of a large proportion of investor portfolios' stakes being committed to social impact investments (Yu and Zhao 2015). The same authors established that it is desirable to assess the impact of the sustainability capital market because the DJSI is a global index and firms are evaluated worldwide. Additionally, the work of the above authors incorporated the antecedents of the status in Asian emerging countries in which companies that are incorporated into the DJSI have a higher market valuation than companies that have not been chosen to be part of this index.

Lameira et al. (2013) found evidence to suggest that the higher the company value is, the greater that the probability is that it participates in the sustainability index and the more likely that it is to demonstrate better sustainability practices. Conversely, these companies' return on assets will be higher, and the probability that they exhibit optimal sustainability practices will be lower.

Regarding leverage, Charlo et al. (2015) indicate that companies with a responsibility index show the highest levels of external financing. The positive relationship between responsible companies and leverage appears to confirm that this type of business has the greatest ability to prove that social responsibility is positively valued by investment funds and other investors. The results show that socially responsible Spanish companies tend to be larger than companies that are not responsible and that the former provide larger returns on market risk.

In their study, Leuz et al. (2003) note that larger companies have higher revenue and that leverage for operations is positively related to income volatility. Similarly, inflation rates and growth index volatility influence accounting income variability. Additionally, in their investigation, Zhao and Murrell (2016) reported that CSP might not have a positive influence on CFP.

As Charlo et al. (2015) argued, businesses have no responsibility other than achieving the highest possible profits, so investing in CSR implies an additional cost that places a company at a disadvantage in relation to its competitors because of less net income. However, complementing the statement by Waddock and Graves (1997) that good financial results allow money to be invested in CSR; hence, high profits could be a good indicator of the subsequent results of social investments (Barnett 2007). Charlo et al. (2015) indicate that socially responsible companies obtain higher profits for the same systematic risk as companies that are not socially responsible.

Hansen and Schaltegger (2018) indicate that organizational structures aim to achieve sustainability objectives and not to maximize financial gains since structures or models subordinate social and environmental publications to economic objectives.

## 3. Materials and Methods

The theoretical foundation of this study was developed by verifying the most relevant categories on which this research focuses, which are associated with indices and variables linked to business sustainability, considering the DJSI Chile is similar to FTSE4GOOD (Montoya-Cruz et al. 2020) in that the inclusion of a company in the index is based on a range of corporate social responsibility criteria, the index is designed to measure the performance of companies demonstrating strong Environmental, Social, and Governance (ESG) practices. The DJSI Chile is a float-adjusted market

capitalization-weighted index that measures the performance of the companies in the Bolsa de Santiago's IGPA Index, (the underlying index universe) that meet minimum sustainability requirements, as defined in ESG (Environmental, Social, Governance) criteria using a best-in-class approach.

The definitions of the variables are presented in Appendix A based on the theoretical foundation of the research.

The research methodology was quantitative, and the following hypotheses were proposed:

**Hypothesis 1a (H1a).** *The financial indices of companies are correlated with the DJSI Chile, especially the companies that are part of that index.*

**Hypothesis 1b (H1b).** *The companies that are part of the DJSI Chile have a greater correlation with the financial indices than the companies that are part of the IGPA and that are not considered in the DJSI Chile because the former have a greater investment in and concern for sustainability.*

Accepting the research hypotheses would indicate that the DJSI is an indicator of the financial projections of the most sustainable Chilean companies, differentiating them from other companies belonging to the IGPA in Chile.

The IGPA Chile, the index seeks to measure the performance of Chile domiciled stocks listed on the Santiago Exchange that have a presence greater than or equal to 25% as well as meeting other minimum liquidity criteria, the stocks must have an Investable Weight Factor greater than or equal to 5%, presence must be greater than or equal to 25%, and the stocks must have an annual traded value in excess of Unidades de Fomento (equal to 37 Dollar American), 10.000 as of the rebalancing reference date.

The background for the study was based on documents published by the companies, mainly their annual reports and information in the Bloomberg database.

For the quantitative analysis, all 83 companies contained in the IGPA through September 2018 were considered, 29 of which belonged to the DJSI Chile.

Financial information for the 83 companies that make up the IGPA for the period 2015–2018 was obtained, considering that the index began operating in June 2015.

The information capture was organized considering two variables for comparison the correlation with DJSI Chile with the finance index (Ramos-Requena et al. 2020). It was to apply the Shapiro–Wilk normality test, which was chosen because of the amount of data available for the variables ROA, ROE, market value, earnings, and leverage as of 31 December 2015, 31 December 2016, 31 December 2017, and 31 December 2018. Then, if the data presented corresponded to a parametric test, Pearson's correlation test was used, and if the data corresponded to a nonparametric test, Spearman's rho correlation test was used.

## 4. Results

To obtain the correlations of the variables presented in the investigation, the normality tests shown in Table 1 were performed and presented. The normal test used for the investigation was the Shapiro–Wilk (W) test, and the Prob < W value listed in the output was the *p*-value. If the alpha level was 0.05 and the *p*-value was less than 0.05, then the null hypothesis that the data were normally distributed was rejected.

In the analysis, the Shapiro–Wilk test was used to determine the normal distribution of the data for the companies under study, and Pearson's correlation test and Spearman's rho test were applied in cases of parametric and nonparametric distribution, respectively. Parametric tests were used for variables with large amounts of data, and, with exceptions, when nonparametric data were available; both tests were applied here to eliminate possible errors in the results. The correlation test results are shown in Appendix B.

**Table 1.** Percentage normality tests according to the financial indices of the companies under study and that belong to the Dow Jones Sustainability Index (DJSI).

| Parametric and No Parametric in Function the Shapiro-Wilk | Shapiro–Wilk ROE | Shapiro–Wilk ROA | Shapiro–Wilk Market Value | Shapiro–Wilk Earnings | Shapiro–Wilk Leverage |
|---|---|---|---|---|---|
| Parametric belong to the DJSI | 93.10% | 82.76% | 96.55% | 96.55% | 96.55% |
| Nonparametric belong to the DJSI | 6.90% | 17.24% | 3.45% | 3.45% | 3.45% |
| Parametric not belong to the DJSI | 77.42% | 79.03% | 74.19% | 83.87% | 88.71% |
| Nonparametric not belong to the DJSI | 22.58% | 20.97% | 25.81% | 16.13% | 11.29% |

Source: Self-made based on data Bloomberg data.

The model for the correlation test is as follows.

For the estimation of the model in Equations (1) and (2), the following functional form is assumed for the Pearson's correlation test:

$$r = \frac{n(\sum xy) - (\sum x)(\sum y)}{\sqrt{\left[n\sum x^2 - (\sum x)^2\right]\left[n\sum y^2 - (\sum y)^2\right]}}, \tag{1}$$

where:

r: Pearson's correlation

n: size, for variable the DJSI indices and the finance variable

x, y: variable considering the year, and considering the variable DJSI Chile and finance index. For Spearman's rho correlation test (Rho)

$$r_s = \frac{\mathrm{cov}\left(rg_x, rg_y\right)}{\theta rg_x\, \theta rg_y}, \tag{2}$$

where:

$r_s$: denotes the usual Pearson's correlation coefficient, but applied to the rank variables, include DJSI Chile and finance index.

$\mathrm{cov}\left(rg_x, rg_y\right)$: is the covariance of the rank variables and

$\theta rg_x\, \theta rg_y$: are the standard deviations of the rank variables.

Table 2 shows the percentages of companies that demonstrate a positive and negative correlation, including the companies that are part of the DJSI and those that are part of the IGPA but not part of the DJSI Chile, in relation to ROE.

**Table 2.** Summary of ROE correlations with the DJSI Chile.

| Companies Make up DJSI Chile or IGPA | Positive Correlation | | No Correlation | | Negative Correlation | | Total |
|---|---|---|---|---|---|---|---|
| | Pearson's Correlation | Rho's Correlation | Pearson's Correlation | Rho's Correlation | Pearson's Correlation | Rho's Correlation | |
| Companies that make up the DJSI Chile | 29% | 39% | 32% | 39% | 39% | 22% | 100% |
| Companies that make up the IGPA | 41% | 43% | 38% | 43% | 21% | 14% | 100% |

Source: Self-made.

Table 2 shows that there is no correlation between ROE and the DJSI Chile in most of the companies analyzed, and the companies that are part of the DJSI do not exhibit a stronger relationship with ROE.

According to Table 3, there is a correlation between the DJSI and ROA for companies that are part of the DJSI and not part of the DJSI Chile but belonging to the IGPA Chile. In addition, compared to companies that are part of the DJSI Chile, the result of the companies that are and are not part of the DJSI Chile show a positive correlation with ROA.

**Table 3.** Summary of ROA correlations with the DJSI Chile.

| Companies Make up DJSI Chile or IGPA | Positive Correlation | | No Correlation | | Negative Correlation | | Total |
|---|---|---|---|---|---|---|---|
| | Pearson's Correlation | Rho's Correlation | Pearson's Correlation | Rho's Correlation | Pearson's Correlation | Rho's Correlation | |
| Companies that make up the DJSI Chile | 32% | 39% | 36% | 35% | 32% | 26% | 100% |
| Companies that make up the IGPA | 40% | 38% | 36% | 46% | 24% | 16% | 100% |

Source: Self-made.

Table 4 indicates that the majority of companies (approximately 92%) maintain a positive correlation with market value regardless of whether they belong to the DJSI Chile.

**Table 4.** Summary of market value correlation with the DJSI Chile.

| Companies Make up DJSI Chile or IGPA | Positive Correlation | | No Correlation | | Negative Correlation | | Total |
|---|---|---|---|---|---|---|---|
| | Pearson's Correlation | Rho's Correlation | Pearson's Correlation | Rho's Correlation | Pearson's Correlation | Rho's Correlation | |
| Companies that make up the DJSI Chile | 84% | 84% | 13% | 16% | 3% | 0% | 100% |
| Companies that make up the IGPA | 92% | 91% | 5% | 7% | 3% | 2% | 100% |

Source: Self-made.

Table 5 indicates that more than 50% of the DJSI Chile companies have a positive correlation with earnings. It is possible to conclude that earnings are correlated with belonging to the DJSI Chile in comparison with other companies that are part of the IGPA.

**Table 5.** Summary of earnings correlation with the DJSI Chile.

| Companies Make up DJSI Chile or IGPA | Positive Correlation | | No Correlation | | Negative Correlation | | Total |
|---|---|---|---|---|---|---|---|
| | Pearson's Correlation | Rho's Correlation | Pearson's Correlation | Rho's Correlation | Pearson's Correlation | Rho's Correlation | |
| Companies that make up the DJSI Chile | 48% | 58% | 36% | 32% | 16% | 10% | 100% |
| Companies that make up the IGPA | 45% | 41% | 36% | 47% | 19% | 12% | 100% |

Source: Self-made.

Table 6 shows the number of companies correlated with leverage and even shows a balanced correlation among negative, positive, and no correlation independence regardless of whether the companies belong to the DJSI Chile.

**Table 6.** Summary of leverage correlation with the DJSI Chile.

| Companies Make up DJSI Chile or IGPA | Positive Correlation | | No Correlation | | Negative Correlation | | Total |
|---|---|---|---|---|---|---|---|
| | Pearson's Correlation | Rho's Correlation | Pearson's Correlation | Rho's Correlation | Pearson's Correlation | Rho's Correlation | |
| Companies that make up the DJSI Chile | 29% | 42% | 13% | 19% | 58% | 39% | 100% |
| Companies that make up the IGPA | 31% | 28% | 33% | 40% | 36% | 32% | 100% |

Source: Self-made.

For the first hypothesis, H1a, most of the companies have a positive correlation with the DJSI Chile for the market value variable.

With regard to the second scenario, H1b the earnings-related variable has a slightly higher correlation for the companies that make up the DJSI Chile than for the rest of the companies in the IGPA.

## 5. Discussion

Considering that sustainable development has been implemented for the management of problems associated with global warming, natural resources, and responsibility for income generation and that, for their part, companies in emerging countries, especially Chile, have focused on four main aspects of sustainable development—the environment, the market, corporate governance, and relationship with the community—it is important to link the index associated with sustainability to the economic success of companies.

According to Charlo et al. (2015), Chen et al. (2018), and Lameira et al. (2013), for the same level of risk, different benefits arise from the implementation of sustainable practices by companies; for example, there is a slightly higher correlation for earnings among companies that belong to the DJSI Chile than among companies that belong to the IGPA. Charlo et al. (2015) together with Hoi et al. (2018) indicate that leverage rates and changes in the market are also affected through share value. In this investigation, there is a positive relationship between the sample indices, including companies that have a positive correlation with the index, both companies that make up the index as well as those that are part of the IGPA. In the case of market value, a high percentage of companies link their market value with the DJSI index, but there is no significant difference in terms of whether the company is part of the DJSI index or the IGPA.

In the case of ROE and in addition to all of the previous finance variables, Biasin et al. (2019) indicate that there is a positive relationship between the implementation of sustainable practices and all company finance variables, such as ROA, ROE, earnings per share, market value, share value, and EBITDA. ROE was found to be positively correlated with the index in 39% of the companies that are part of the DJSI and in 43% of the companies that are part of the general IGPA.

Therefore, for the four years during which this sustainability index has been in use, the first hypothesis, namely, that market value has a positive correlation with the DJSI Chile, is accepted. For the second hypothesis, there are no significant differences with the exception of the earnings variable in terms of whether belonging to the DJSI Chile has a positive correlation with the DJSI. This finding is based on the work of Lameira et al. (2013), who observed that it is impossible to indicate the result for each variable since studies have employed variables with different effects. In this case, market value is not necessarily based on sustainable practices; therefore, the correlation results for the variables are inconclusive. For earnings, however, the correlation is slightly higher for companies that make up the DJSI Chile than for the rest of the companies that belong to the IGPA.

## 6. Conclusions

Regarding the results for the ROE, ROA, and leverage variables, there is no positive correlation with the DJSI Chile, and being included in an index associated with sustainability does not indicate a greater correlation. Thus, during the four years of the study period, the inclusion of a company in the index does not result in a greater correlation with ROA, ROE, or leverage. The conclusion is different for market value and earnings, however, since market value has a positive correlation for companies belonging to the DJSI Chile compared to the remaining companies that belong to the IGPA, and for the earnings variable, the correlation is slightly higher for companies that are part of the DJSI Chile than for other companies, by the index finances (value market and earnings), it is possible to realize a prediction about the DJSI Chile.

The companies from developed countries that are part of the DJSI, which include companies across all industries, outperform their counterparts in terms of sustainability measures. It is also important to incorporate finances indexes related to sustainability to examine correlations, such as environmental investment, internal and external social investments, and investments in corporate governance policies because socially responsible financial investment has taken on particular importance and investors select the most profitable investments and have appreciated that companies develop socially responsible policies.

A limitation of this study regarding the degree of correlation is the number of years of operation of the companies because the index began to be used in June 2015 in Chile. Another limitation is associated with the incorporated variables linked with new product development associated with the circular economy for the competitive advantage and financial return. For future research, it is important to consider earnings and market value for this relationship with the sustainability index with the possibility of predictive models associated with finance return or behavior in the stock market.

**Funding:** This research received no external funding

**Conflicts of Interest:** The authors declare no conflict of interest.

## Appendix A

**Table A1.** Variables incorporated in this research according to the authors.

| Variable | Explanation | Index | Referenced Authors |
|---|---|---|---|
| ROA | Return on assets | Profitability/total assets | Yu and Zhao (2015) |
| ROE | Return on equity | Profitability/equity | Lameira et al. (2013); Hoi et al. (2018) |
| Earnings | Earnings before interest, taxes, depreciation, and amortization | EBITDA | Lameira et al. (2013) |
| Leverage | Debt with third parties based on the investment made by the shareholders of the companies under study | Debt with third parties/total liabilities plus equity | Lameira et al. (2013) |
| Market value | Value of the share in the Santiago Stock Exchange | Value of the share in the market | Yu and Zhao (2015); Oikonomou et al. (2018); Ioannou and Serafeim (2015) |
| REPU | 40% of companies with the best sustainability index, incorporated into the DJSI Chile | Incorporation in the DJSI Chile | Charlo et al. (2015); Yu and Zhao (2015); Lo and Sheu (2007); Robinson et al. (2011); Saeidi et al. (2015). |

Source: Self-made.

## Appendix B

**Table A2.** Correlation of company variables with the DJSI.

| | ROE | | ROA | | Market Value | | Earnings | | Leverage | |
|---|---|---|---|---|---|---|---|---|---|---|
| | Pearson | Rho | Pearson | Rho | Pearson | Rho | Pearson | Rho | Pearson | Rho |
| Agunsa | −0.645 | −0.200 | −0.640 | −0.200 | 0.746 | 0.800 | −0.715 | −0.200 | −0.429 | −0.400 |
| Almendral | 0.247 | 0.400 | 0.244 | 0.400 | 0.249 | 0.400 | 0.181 | 0.400 | −0.601 | −0.400 |
| Antarchile | 0.394 | 0.600 | 0.436 | 0.738 | 0.962 * | 1.000 ** | 0.392 | 0.600 | 0.340 | 0.400 |
| Australis | 0.791 | 0.800 | 0.879 | 0.800 | 0.837 | 0.800 | 0.868 | 1.000 ** | −0.873 | −1.000 ** |
| Banvida | 0.264 | 0.400 | 0.292 | 0.400 | 0.998 ** | 1.000 ** | 0.503 | 0.400 | −0.667 | −0.800 |
| Besalco | −0.618 | −0.800 | −0.711 | −0.632 | 0.951 * | 1.000 ** | −0.636 | −0.800 | −0.361 | −0.400 |
| Colocolo | 0.055 | −0.200 | 0.127 | −0.200 | 0.671 | 0.600 | 0.096 | −0.200 | 0.904 | 1.000 ** |
| Blumar | 0.468 | 0.400 | 0.457 | 0.400 | 0.630 | 0.800 | 0.455 | 0.400 | −0.522 | −0.400 |
| Cementos | 0.042 | 0.200 | 0.698 | 0.632 | 0.891 | 0.800 | 0.502 | 0.400 | −0.933 | −0.800 |
| Cintac | 0.821 | 0.800 | 0.827 | 0.800 | 0.888 | 0.800 | 0.759 | 0.800 | −0.023 | 0.000 |
| Las Condes | −0.823 | −0.800 | −0.876 | −0.800 | −0.612 | −0.800 | −0.828 | −0.800 | 0.858 | 0.800 |
| Embonorb | 0.818 | 1.000 ** | 0.733 | 0.400 | 0.966 * | 1.000 ** | 0.825 | 0.800 | 0.715 | 0.600 |
| Ccu | 0.336 | 0.600 | 0.289 | −0.211 | 0.852 | 0.800 | 0.355 | 0.600 | 0.635 | 0.800 |
| Interocean | −0.745 | −0.200 | −0.749 | −0.200 | 0.264 | 0.000 | −0.753 | −0.200 | 0.753 | 0.800 |
| Camanchaca | 0.715 | 0.800 | 0.643 | 0.400 | 0.753 | 0.800 | 0.714 | 0.800 | −0.330 | −0.400 |
| Vapores | −0.627 | −0.400 | −0.627 | −0.400 | 0.909 | 1.000 ** | −0.613 | −0.400 | −0.791 | −0.800 |
| Cristales | −0.363 | −0.316 | −0.405 | −0.600 | 0.722 | 0.400 | −0.066 | 0.000 | 0.599 | 0.400 |
| Eisa | −0.647 | −0.400 | −0.690 | −0.400 | 0.988 * | 1.000 ** | −0.440 | −0.400 | 0.866 | 0.800 |
| Moller | 0.296 | 0.400 | −0.132 | 0.200 | 0.738 | 0.800 | 0.327 | 0.400 | 0.694 | 0.800 |
| Pehuenche | −0.896 | −0.800 | −0.845 | −0.800 | −0.987 * | −1.000 ** | −0.865 | −0.800 | −0.915 | −1.000 ** |
| Aquachile | 0.906 | 1.000 ** | 0.928 | 1.000 ** | 0.662 | 0.800 | 0.898 | 0.800 | −0.367 | −0.400 |
| Hites | −0.485 | −0.600 | −0.519 | −0.600 | 0.905 | 0.800 | −0.319 | 0.000 | 0.526 | 0.600 |
| Nuevapolar | 0.749 | 0.800 | 0.760 | 0.800 | 0.827 | 1.000 ** | 0.753 | 0.800 | 0.213 | −0.400 |
| Lipigas | 0.435 | 0.400 | −0.051 | 0.200 | 0.814 | 0.800 | 0.993 ** | 1.000 ** | 0.705 | 0.400 |
| Tricot | 0.939 | 0.949 | 0.946 | 0.949 | 0.881 | 0.738 | 0.934 | 0.949 | 0.872 | 0.738 |
| Enelchile | −0.774 | −0.400 | −0.773 | −0.400 | −0.590 | −0.200 | 0.822 | 0.800 | −0.456 | −0.800 |
| Enelgxch | −0.118 | 0.000 | 0.093 | 0.211 | 0.966 * | 1.000 ** | −0.218 | 0.000 | −0.995 ** | −1.000 ** |
| Ecl | −0.309 | −0.600 | −0.265 | 0.316 | 0.975 * | 1.000 ** | −0.301 | −0.400 | −0.843 | −0.800 |
| Enjoy | 0.114 | −0.200 | 0.002 | −0.200 | 0.911 | 0.800 | −0.039 | −0.200 | 0.621 | 0.400 |
| Edelpa | 0.926 | 0.800 | 0.932 | 0.800 | 0.955 * | 1.000 ** | 0.929 | 0.800 | −0.923 | −1.000 ** |
| Pasur | 0.544 | 0.600 | 0.544 | 0.600 | 0.620 | 0.800 | 0.519 | 0.600 | −0.016 | 0.200 |
| Forus | −0.820 | −0.800 | −0.831 | −0.800 | 0.815 | 0.800 | −0.756 | −0.400 | −0.560 | −0.400 |

**Table A2.** *Cont.*

| | ROE | | ROA | | Market Value | | Earnings | | Leverage | |
|---|---|---|---|---|---|---|---|---|---|---|
| | Pearson | Rho | Pearson | Rho | Pearson | Rho | Pearson | Rho | Pearson | Rho |
| Gasco | 0.315 | 0.400 | −0.172 | 0.200 | 0.751 | 0.800 | −0.678 | −0.400 | 0.543 | 0.200 |
| Hf | −0.012 | −0.200 | −0.100 | −0.400 | 0.949 | 0.800 | 0.343 | 0.400 | 0.549 | 0.400 |
| Ingevec | 0.917 | 0.949 | 0.989 * | 1.000 ** | 0.896 | 0.800 | 0.822 | 0.800 | −0.770 | −0.800 |
| Indisa | −0.984 * | −1.00 ** | −0.825 | −0.800 | 0.914 | 1.000 ** | −0.165 | −0.400 | −0.375 | −0.600 |
| Invercap | 0.935 | 0.800 | 0.935 | 0.800 | 0.993 ** | 1.000 ** | 0.943 | 0.800 | −0.672 | −0.800 |
| Invermar | −0.693 | −0.200 | −0.457 | −0.200 | 0.915 | 0.800 | −0.427 | −0.200 | −0.832 | −0.800 |
| Ilc | −0.204 | 0.000 | −0.281 | 0.000 | 0.951 * | 0.800 | −0.056 | 0.400 | 0.665 | 0.800 |
| Masisa | −0.979 * | −1.00 ** | −0.975 * | −1.000 ** | 0.897 | 0.800 | −0.938 | −0.800 | −0.876 | −0.800 |
| Minera | 0.844 | 0.600 | 0.887 | 0.800 | 0.611 | 0.600 | 0.612 | 0.600 | −0.894 | −0.800 |
| Molymet | −0.058 | 0.200 | 0.147 | 0.200 | 0.955 * | 0.800 | 0.070 | 0.200 | −0.728 | −0.800 |
| Multifoods | 0.860 | 1.000 ** | 0.907 | 1.000 ** | 0.735 | 0.800 | 0.888 | 0.800 | −0.936 | −0.800 |
| Nitratos | 0.896 | 0.800 | 0.872 | 0.800 | 0.313 | 0.800 | 0.828 | 0.800 | −0.820 | −0.800 |
| Nortegran | 0.917 | 0.800 | 0.957 * | 1.000 ** | 0.967 * | 1.000 ** | 0.850 | 0.800 | −0.601 | −0.800 |
| Pacifico | 0.338 | 0.800 | 0.332 | 0.800 | 0.776 | 1.000 ** | 0.299 | 0.400 | 0.319 | 0.400 |
| Paz | −0.726 | −0.400 | −0.721 | −0.400 | 0.946 | 0.800 | −0.704 | −0.400 | −0.233 | 0.000 |
| Quinenco | −0.274 | −0.400 | −0.387 | −0.400 | 0.997 ** | 1.000 ** | 0.141 | 0.400 | −0.438 | −0.400 |
| Ripley | 0.609 | 0.400 | 0.612 | 0.400 | 0.995 ** | 1.000 ** | 0.630 | 0.400 | −0.858 | −0.800 |
| Salfacorp | −0.963 * | −0.949 | −0.576 | −0.600 | 0.976 * | 1.000 ** | 0.666 | 0.400 | −0.039 | 0.000 |
| Smchileb | 0.309 | −0.400 | 0.312 | 0.258 | 0.946 | 0.800 | −0.420 | −0.800 | −0.396 | −0.800 |
| Smchiled | 0.309 | −0.400 | 0.312 | 0.258 | 0.956 * | 0.800 | −0.420 | −0.800 | −0.396 | −0.800 |
| Smu | 0.815 | 0.800 | 0.874 | 0.800 | 0.929 | 0.800 | 0.879 | 0.800 | −0.867 | −0.800 |
| Oroblanco | 0.943 | 0.949 | 0.972 * | 1.000 ** | 0.949 | 1.000 ** | 0.864 | 0.800 | 0.005 | -0.400 |
| Coloso | −0.670 | −0.800 | −0.628 | −0.800 | 0.817 | 0.800 | −0.755 | −0.800 | 0.768 | 1.000 ** |
| Pucobre | 0.940 | 1.000 ** | 0.922 | 0.800 | 0.991 ** | 1.000 ** | 0.954 * | 1.000 ** | 0.604 | 0.800 |
| Sqma | 0.923 | 0.800 | 0.947 | 0.800 | 0.929 | 0.800 | 0.863 | 0.800 | 0.232 | 0.000 |
| Sqmb | 0.923 | 0.800 | 0.947 | 0.800 | 0.997 ** | 1.000 ** | 0.863 | 0.800 | 0.232 | 0.000 |
| Socovesa | −0.073 | 0.000 | 0.400 | 0.447 | 0.953 * | 0.800 | 0.962 * | 1.000 ** | −0.999 ** | −1.000 ** |
| Soquicom | −0.934 | −1.00 ** | −0.953 * | −1.000 ** | 0.969 * | 1.000 ** | −0.961 * | −1.000 ** | 0.989 * | 1.000 ** |
| Watts | −0.973 * | −1.00 ** | −0.980 * | −1.000 ** | 0.624 | 0.400 | −0.998 ** | −1.000 ** | −0.236 | −0.200 |
| Zofri | 0.557 | 0.600 | 0.602 | 0.600 | 0.846 | 0.800 | 0.633 | 0.800 | −0.659 | −0.600 |
| Habitat | −0.256 | −0.105 | −0.329 | −0.400 | 0.902 | 0.800 | 0.925 | 1.000 ** | 0.885 | 0.800 |
| Aesgener | −0.668 | −0.400 | −0.503 | −0.400 | −0.878 | −1.000 ** | −0.621 | −0.400 | −0.669 | −0.800 |
| AguasA | 0.026 | 0.400 | −0.226 | 0.000 | 0.927 | 1.000 ** | 0.222 | 0.400 | 0.533 | 0.800 |
| Chile | −0.868 | −0.800 | −0.025 | 0.000 | 0.943 | 0.800 | 0.677 | 0.600 | −0.946 | −1.000 ** |

**Table A2.** *Cont.*

| | ROE | | ROA | | Market Value | | Earnings | | Leverage | |
|---|---|---|---|---|---|---|---|---|---|---|
| | **Pearson** | **Rho** | **Pearson** | **Rho** | **Pearson** | **Rho** | **Pearson** | **Rho** | **Pearson** | **Rho** |
| Bci | −0.693 | −0.400 | −0.312 | −0.258 | 0.902 | 0.800 | 0.805 | 0.800 | −0.740 | −0.400 |
| Itaucorp | −0.668 | −0.400 | −0.560 | −0.400 | 0.508 | 0.800 | −0.379 | −0.400 | −0.802 | −1.000 ** |
| Bsantander | 0.949 | 0.949 | 0.954 * | 0.949 | 0.887 | 0.800 | 0.895 | 0.800 | −0.889 | −0.800 |
| Cap | 0.978 * | 1.000 ** | 0.970 * | 1.000 ** | 0.989 * | 1.000 ** | 0.940 | 0.800 | −0.727 | −0.800 |
| Cencosud | 0.396 | 0.400 | 0.419 | 0.400 | 0.310 | 0.000 | 0.410 | 0.400 | −0.867 | −0.800 |
| Colbun | 0.874 | 0.800 | 0.948 | 1.000 ** | −0.094 | −0.400 | 0.804 | 0.600 | −0.891 | −1.000 ** |
| AndinaA | 0.868 | 1.000 ** | 0.877 | 1.000 ** | 0.963 * | 1.000 ** | 0.895 | 1.000 ** | −0.240 | 0.000 |
| AndinaB | 0.868 | 1.000 ** | 0.877 | 1.000 ** | 0.991 ** | 1.000 ** | 0.895 | 1.000 ** | −0.240 | 0.000 |
| Entel | 0.287 | 0.400 | 0.250 | 0.400 | 0.047 | 0.000 | 0.259 | 0.400 | −0.670 | −0.400 |
| Cmpc | 0.498 | 0.600 | 0.492 | 0.600 | 0.806 | 0.600 | 0.477 | 0.600 | −0.572 | −0.738 |
| Copec | 0.357 | 0.600 | 0.365 | 0.600 | 0.980 * | 1.000 ** | 0.345 | 0.600 | 0.356 | 0.400 |
| Enaex | −0.888 | −0.800 | −0.903 | −0.800 | 0.909 | 0.800 | −0.918 | −1.000 ** | −0.875 | −1.000 ** |
| Enelam | 0.279 | 0.000 | −0.895 | −0.949 | 0.986 * | 1.000 ** | −0.083 | 0.000 | 0.684 | 0.800 |
| Security | −0.806 | −0.800 | −0.745 | −0.775 | 0.915 | 0.800 | 0.730 | 0.800 | −0.891 | −0.800 |
| Iam | 0.088 | 0.400 | −0.203 | 0.000 | 0.975 * | 1.000 ** | 0.233 | 0.400 | 0.567 | 0.800 |
| Ltm | 0.863 | 0.800 | 0.884 | 0.800 | 0.993 ** | 1.000 ** | 0.860 | 0.800 | −0.818 | −0.800 |
| Parauco | 0.815 | 0.800 | 0.763 | 0.800 | 0.970 * | 1.000 ** | 0.723 | 0.800 | 0.971 * | 1.000 ** |
| Falabella | −0.757 | −0.600 | −0.724 | −0.600 | 0.907 | 0.800 | −0.410 | −0.600 | −0.813 | −0.800 |
| Sk | −0.800 | −0.949 | −0.774 | −0.775 | 0.969 * | 1.000 ** | −0.824 | −0.800 | −0.857 | −0.800 |
| Smsaam | −0.720 | −0.400 | −0.834 | −0.800 | 0.991 ** | 1.000 ** | −0.741 | −0.400 | 0.940 | 0.800 |
| Sonda | 0.162 | 0.200 | 0.114 | 0.200 | −0.071 | 0.200 | 0.238 | 0.200 | 0.950 * | 0.800 |
| Concha y Toro | −0.839 | −0.800 | −0.748 | −0.800 | 0.624 | 0.800 | 0.089 | −0.200 | −0.823 | −0.800 |
| Vspt | −0.832 | −0.600 | −0.799 | −0.600 | 0.772 | 0.800 | −0.805 | −0.600 | −0.820 | −0.800 |

Source: Self-made. This table shows the correlation of the financial index with the DJSI applying Pearson's test and Spearman's rho test. * La correlación es significativa en el nivel 0.05 (bilateral). ** La correlación es significativa en el nivel 0.01 (bilateral).

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
