# Peer review of "Correlation between the DJSI Chile and the Financial Indices of Chilean Companies"

_ijfs, doi:10.3390/ijfs8040074_

Round 1

Reviewer 1 Report

Attached I send the review

Author Response

  1. I would encourage the authors to extend the abstract more with the key results. The abstract was extended including the conclusions of all the variables studied.

Response:

The abstract was adjusted and shown below

However, the DJSI Chile is correlated with market value (for approximately eighty percent of the companies), and with earnings, there is a slightly higher correlation for the companies that belong to the DJSI Chile than for the remaining companies in the IGPA, thus if exist the correlation between DJSI Chile index with variables market value and earnings, the index enables the prediction of those financial variables or predict the DJSI Chile in the function the finance indices (value market and earnings) of the companies the make up the DJSI Chile.

  1. I would advise that Tables 2 and 8 be included as an appendix.

Response:

The Table 2 was included in Appendix 1.

The Table 8 was included in Appendix 2.

Change of the number tables

  1. I would recommend that you include in your references a recent paper: Ramos-Requena, J. P., Trinidad-Segovia, J. E., & Sánchez-Granero, M. Á. (2020). An Alternative Approach to Measure Co-Movement between Two Time Series. Mathematics, 8(2), 261 and Montoya-Cruz, E., Ramos-Requena, J. P., Trinidad-Segovia, J. E., & Sánchez-Granero, M. Á. (2020). Exploring Arbitrage Strategies in Corporate Social Responsibility Companies. Sustainability, 12(16), 6293.

Response:

It was incorporated the reference for the method (Ramos-Requena, Trinidad-Segovia & Sánchez-Granero, 2020). Line 222-223.

It was incorporated the reference (Montoya-Cruz, Ramos-Requena, Trinidad-Segovia & Sánchez-Granero, 2020), in the line 190-191.

Reviewer 2 Report

-it is an interesting work but it needs improvements - the introduction from a quantitative point of view is sufficient but as a content it does not have a logical succession of ideas, it does not have a "red thread of presentation" - going through the paper, only general conclusions can be deduced, for example Line 141-143, probably on short ground… .. - the material and the method used in the research I think should express more clearly the objective pursued and be presented before the results - I don't think it is necessary to list all the names of the companies, maybe they should be divided into size and activity classes - after reading the paper no clear conclusions can be drawn - I consider that a major revision of the structure and content requires some mistakes, for example Row 283

Author Response

  1. The introduction from a quantitative point of view is sufficient but as a content, it does not have a logical succession of ideas.

Response:

The introduction was made shorter and the section of the literature review was added, that recommended review 3

  1. It does not have a "red thread of presentation" going through the paper, only general conclusions can be deduced, for example, Line 141-143, probably on short ground…

Response:

Adjusted in the literature review (line 176-178) “because of less net income. However, complementing the statement by Waddock and Graves (1997)”  

  1. The material and the method used in the research I think should express more clearly the objective pursued and be presented before the results.

Response:

The Material and the method used were organized and change after the introduction and before the Results.

  1. I don't think it is necessary to list all the names of the companies, maybe they should be divided into size and activity classes - after reading the paper no clear conclusions can be drawn.

Response:

Table 2 changes for Appendix 1, but is possible characterizer the companies to the IGPA and DJSI Chile, this index was characterizer in the function of companies in line 209 at 214, and 221 at 223, considerer the stock index associated with leading sustainability companies.

  1. I consider that a major revision of the structure and content requires some mistakes, for example, Row 283

Response:

Was eliminated the use of ATLAS.ti software, for possible confusion about the review.

Reviewer 3 Report

The paper examines the correlation between the companies in the Dower Jones Sustainability Chile Index (DJSI) with the financial indexes of the sample companies in the General Stock Price Index (IGPA) in Chile. The research includes an analysis based on the Shapiro-Wilk normality test and a correlation test. The author(s) test two hypotheses relating to, (a) financial indexes of the companies are correlated in the DJSI Chile, and (b) companies that are part of the DJSI Chile have a greater correlation with the financial indexes than the companies that are part of the IGPA and are not considered in the DJSI Chile.

There is no doubt that the current literature regarding a correlation between improved firm performance and the adoption of sustainability practices is not conclusive in its findings.

Detailed comments:

  1. The layout of the paper should be rearranged so that the Methods section comes before the ’Results’.
  2. The methodology is appropriate for the size of the sample. However, a possible limitation of the study is that it does not take into account strategic variables such as investment in new product development (NPD), which has been shown to enhance competitive advantage and financial return.
  3. All acronyms should be spelt out first, even well-known ones such as ROA and ROE.
  4. Table 2 would be better as an appendix to the paper.
  5. The formula for Pearson correlation looks as though it has been cut and pasted. The same applies to Spearman’s rho correlation test. See also lines 202 – 213 for errors.
  6. The literature review (mainly in the Introduction) is appropriate. It may, however, be better to have a shorter ‘Introduction’, followed by a ‘Literature Review’.
  7. There are a few English grammar improvements that could be made to the paper. For example - Line 35. ‘results; however, it’, should be ‘results. However, it’.
  8. Long sentences should be avoided. See for example Line 25 to Line 29. The sentence should be broken at Line 26 ‘… activities; this’ - … activities. This’. Also, Line 47 to Line 52. In some cases, instead if using a semicolon (;) a full-stop would be better, followed by the start of a new sentence (see, also, example in comment 7).
  9. The Conclusions section could be improved. In general, the conclusions are supported by the correlation(s) and data analysis. The limitations of the research are highlighted.
  10. Line 414 to Line 419. The reference for Robinson et al. (2011) is entered twice.
  11. The paper, when revised, will contribute to the literature in this area.

Author Response

1. The layout of the paper should be rearranged so that the Methods section comes before the ’Results’.

Response:

The Material and the method used were organized and change after the introduction and before the Results.

2. The methodology is appropriate for the size of the sample. However, a possible limitation of the study is that it does not take into account strategic variables such as investment in new product development (NPD), which has been shown to enhance competitive advantage and financial return.

Response:

The study incorporated strategic variables as the DJSI Chile, in the paper was incorporated this paragraph: Line 194- 198.

It incorporates an index that is the DJSI Chile, which considers companies that invest from a sustainable point of view.

Additional was incorporated the limitation about the incorporated other variables linked with new product development associated with circular economy for the competitive advantage and financial return, line 340- 342.

3. All acronyms should be spelt out first, even well-known ones such as ROA and ROE.

Response:

The acronym was spelt felt ROA and ROE. Line 10 and line 56.

4. Table 2 would be better as an appendix to the paper.

Response:

Table 2 was included in appendix 1

5. The formula for the Pearson correlation looks as though it has been cut and pasted. The same applies to Spearman’s rho correlation test. See also lines 202 – 213 for errors.

Response:

Line 202 at 213 was changed as shown below:

For the estimation of the model in equations (1 and 2), the following functional form is assumed for the Pearson correlation test:

,

(1)

where:

r: Pearson correlation

n: size, for variable DJSI and the finance variable

x, y: variable to consider for year, considerer variable DJSI Chile and finance index. For the Spearman’s rho correlation test (Rho)

 ,

(2)

where:

rs: denotes the usual Pearson correlation coefficient, but applied to the rank variables, include DJSI Chile and finance index.

: is the covariance of the rank variables

: are the standard deviations of the rank variables.

6. The literature review (mainly in the Introduction) is appropriate. It may, however, be better to have a shorter ‘Introduction’, followed by a ‘Literature Review’.

Response:

The introduction was made shorter and divided into introduction and literature review

7. There are a few English grammar improvements that could be made to the paper. For example - Line 35. ‘results; however, it’, should be ‘results. However, it’.

Response:

The line 35 now is line 36, change the “;” for “.”

8. Long sentences should be avoided. See for example Line 25 to Line 29. The sentence should be broken at Line 26 ‘… activities; this’ - … activities. This’. Also, Line 47 to Line 52. In some cases, instead if using a semicolon (;) a full-stop would be better, followed by the start of a new sentence (see, also, example in comment 7).

Response:

The line 26 was adjusted included in line 26. (activities)

      The line 49 now is line 83, the change “;” for “.”

9. The Conclusions section could be improved. In general, the conclusions are supported by the correlation(s) and data analysis. The limitations of the research are highlighted.

Response:

The conclusion was organized and complemented, line 332-345.

10. Line 414 to Line 419. The reference for Robinson et al. (2011) is entered twice.

Response:

The reference Robinson et al (2011) was eliminated because entered twice.

11. The paper, when revised, will contribute to the literature in this area.

Response:

Thank You

Round 2

Reviewer 1 Report

Congratulations

Reviewer 2 Report

Thanks for the changes